# Commercial Pressure Retarded Osmosis Systems for Seawater Desalination Plants

**DOI:** 10.3390/membranes11010069

**Published:** 2021-01-19

**Authors:** Ryo Makabe, Tetsuro Ueyama, Hideyuki Sakai, Akihiko Tanioka

**Affiliations:** 1Kyowakiden Industry Co., Ltd., 10-2 Kawaguchi-Machi, Nagasaki-shi, Nagasaki 852-8108, Japan; ueyama@kyowa-kk.co.jp (T.U.); hidesakai@kyowa-kk.co.jp (H.S.); 2Department of Organic and Polymeric Materials, Tokyo Institute of Technology, 2-12-1 Ookayama, Meguro-ku, Tokyo 152-8552, Japan; atani417@j03.itscom.net

**Keywords:** commercial plant, pressure retarded osmosis, hybrid process, Mega-Ton Water System, seawater contamination

## Abstract

The development of renewable energy technologies is of global importance. To realize a sustainable society, fossil-resource-independent technologies, such as solar- and wind-power generation, should be widely adopted. Pressure retarded osmosis (PRO) is one such potential renewable energy technology. PRO requires salt water and fresh water, both of which can be found at seawater desalination plants. The total power generation capacity of PRO, using concentrated seawater and fresh water, is 3 GW. A large amount of energy is required for seawater desalination; therefore, the introduction of renewable energy should be prioritized. Kyowakiden Industry Co., Ltd., has been working on introducing PRO to seawater desalination plants since 2001 and is attracting attention for its ongoing PRO pilot plant with a scale of 460 m^3^/d, using concentrated seawater and treated sewage water. In this study, we evaluated the feasibility of introducing PRO in existing desalination plants. The feasibility was examined based on technology, operation, and economy. Based on the number of seawater desalination plants in each country and the electricity charges, it was determined whether the introduction of PRO would be viable.

## 1. Introduction

Over the last few decades, petroleum resources have depleted and global warming has increased at alarming rates. To remedy this, energy sources that can facilitate the realization of a sustainable society should be considered [1,2]. Wind, solar, hydroelectric, geothermal, and biomass power generation are gradually becoming widespread [3,4]. Wind power is widely used in deserts, coasts, and offshore. In recent years, photovoltaic power generation has been used by local governments and households, owing to the progressive price reduction. Pressure retarded osmosis (PRO), another useful renewable energy technology, should attract considerable attention in the future [5]. The development and dissemination of useful novel renewable energy sources contribute to the realization of a sustainable society. Kyowakiden Industry Co., Ltd., et al. contributed to the popularization of PRO, which was first proposed by Professor Sidney Loeb in 1976 [5]. PRO is a form of hydroelectric power generation technology; by utilizing the concentration difference, it converts the entropy change into electricity when high- and low-salinity water are mixed [6]. Water with high osmotic pressure includes some factory effluents, salt-lake water, seawater, and concentrated seawater discharged from seawater desalination plants. Most of the concentrated seawater generated from seawater desalination plants is directly discharged into the sea. This concentrated seawater contains high energy potential due to the high entropy but is currently an unused resource.

Similarly, natural seawater has high potential for PRO. However, the performance of commercially available membranes is insufficient to obtain useful energy by PRO with natural seawater and fresh water [7,8,9]. Many researchers have studied PRO with natural seawater and fresh water, to obtain the energy of this unused resource. Membrane manufacturers should develop PRO membranes to realize PRO, using natural seawater and fresh water. In many countries, energy and water shortages are a serious challenge, with 30% of the world’s population living in countries or regions which experience water shortages. These areas often acquire drinking water by using seawater desalination. Seawater desalination has a long history, and there are various technologies [10,11], such as multi-stage flash (MSF), multi-effect desalination (MED), vapor compression, reverse osmosis (RO), and membrane distillation (MD), used to achieve desalination. The energy consumption of seawater desalination depends on seawater desalination technology [12,13,14,15,16]. The RO method, which is used in 69% of seawater desalination plants, consumes 3–4 kWh of energy to for every m^3^ of fresh water produced. The RO method is considered to consume less energy for water production because it does not involve phase changes, as compared to other technologies.

Elimelech et al. evaluated the theoretical minimum energy consumption required for water production as 1.06 kWh/m^3^ [17]. The energy consumption when using a good-quality permeable RO membrane element can be estimated to be 1.8 kWh/m^3^ on a pilot scale. In addition, the energy consumption of current large-scale seawater desalination plants is said to be 3–4 kWh/m^3^ on average [18]. High-performance RO membrane elements are not the only contributors to reducing energy consumption in RO plants. Currently, most commercial RO plants use power-recovery devices to reduce energy consumption, notably, the PX^®^ Pressure Exchanger^®^ by Energy Recovery, Inc. [19]. In addition, Flowserver’s Calder DWEER and Fluid Equipment Development Company’s (FEDCO) Hydraulic pressure boosters (HPB) Turbocharger are used [20]. These are devices for transmitting the pressure of the brine to the feed. In addition, a complex seawater desalination technology called hybrid desalination is being developed as a novel process to reduce the energy consumption of seawater desalination. The development of seawater desalination systems that reduce energy consumption by using multiple processes such as MD–RO, forward osmosis (FO)–RO, and electrodialysis (ED)–RO will continue to be implemented [10,21,22].

Kyowakiden Industry Co., Ltd., et al. proposed PRO for reducing energy consumption in seawater desalination plants [23,24,25,26,27]. The benefits of introducing PRO in seawater desalination plants are not limited to reducing energy consumption. Many seawater desalination plants which are primarily situated in Arab Gulf regions use large amounts of chemicals [28], resulting in marine pollution problems in those regions. To realize a sustainable seawater desalination plant, problems in long-term operation (control of chemical use and solution of marine pollution), solving such problems should be of key importance.

Kyowakiden Industry Co., Ltd., strongly recommends that energy consumption in seawater desalination plants be reduced and measures against marine pollution be implemented by using concentrated seawater, which is an unused resource. In this study, the feasibility of PRO was evaluated from state-of-the-art technological trends (technology), to determine whether PRO can be introduced into existing seawater desalination plants (operation), and the current cost estimation (economic).

## 2. Results and Discussion

### 2.1. State-of-the-Art Technological Trends

#### 2.1.1. PRO Systems

PRO system development is led by Kyowakiden Industry Co., Ltd., in Japan, and Statkraft, in Norway [9,23,24,29]. Kyowakiden Industry Co., Ltd., also conducted the world’s first demonstration test of PRO in seawater desalination plants. In this demonstration test, PRO performance evaluation was conducted by using treated sewage water and concentrated seawater that is discharged into the sea from the sewage-treatment plant and Fukuoka Seawater Desalination Center in Japan, respectively, for over a year. This seawater desalination plant uses a mixed discharge system. In this method, concentrated seawater and treated sewage water are mixed in advance and then discharged into the sea. This discharge method has lower environmental stress on the ocean than other methods. In other words, concentrated seawater and treated sewage water are already mixed at a cost. In this case, energy can be recovered by introducing PRO. Demonstration tests were conducted between 2011 and 2012. Figure 1 shows an image of the demonstration test site. Figure 2 shows the overview of long-term demonstration test site. Table 1 shows materials used at long-term demonstration test site.

The demonstration test site includes eight four-way 10-inch Toyobo cellulose triacetate (CTA) hollow fiber membrane modules, ultrafiltration (UF) membranes, low-pressure RO membranes for advanced treatment of sewage-treated water (pretreatment of feed solution), and a Pelton turbine generator. The hollow fiber membrane has an outer diameter of 0.2 mm, inner diameter of 0.1 mm, and length of 1.3 m. Operating status was acquired by using a 24-hour automatic control and monitoring system. The scale of water volume is 460 and 420 m^3^/d for concentrated seawater and treated sewage, respectively. Through this test, Kyowakiden Industry Co., Ltd., obtained a large amount of power generation (power density: 10 W/m^2^) and temperature changes due to seasonal variation (draw solution (DS), 2–38 °C; feed solution (FS), 1–40 °C) in the performance over the year. Figure 3 shows the time course of permeate flow rate and temperatures at the PRO prototype plant [24]. The permeate flow through the membrane is temperature-dependent, similar to the RO membrane method. The same membrane module was used consecutively in this test. The permeate flow rate did not decrease in the continuous one-year test. At the same time, it participated in the “Mega-ton Water System” organized by Toray Kurihara Fellow. In this project, a 10% energy saving was calculated for a mega-ton-class seawater desalination plant. After introducing PRO, there was an improvement in energy consumption reduction that was proportional to both the concentration of seawater and the amount of concentrated seawater. The PRO system provides excellent performance in large-scale, high-recovery seawater desalination plants.

Statkraft, one of the leading producers of green energy worldwide, has conducted demonstration tests of seawater and river water, and led the development of PRO technology. In long-term demonstration tests, membrane performance monitoring, membrane cleaning and maintenance, PRO PX^®^ operation and pressure control, and freshwater pretreatment optimization were performed. Statkraft estimated the levelized cost of electricity (LCOE) of PRO to be 120 EUR/MWh, which roughly equals to 0.144 USD/kWh (1 EUR = 1.2 USD) at a 25 MW power plant [30]. Kyowakiden Industry Co., Ltd., also announced similar results. However, Statkraft discontinued investment in PRO development, in 2013, due to lack of improvement in membrane performance. The social significance of LCOE published by Statkraft (compared to the electricity bill in the market) is discussed in Section 2.3.

The PRO system has been proposed for various flows. Kyowakiden Industry Co., Ltd., has obtained a patent for the PRO system (PCT/JP2014/051873) [26] that has been registered in Japan, the USA, Australia, Saudi Arabia, and Israel. One of these PRO systems is a method of recovering energy from a turbine. The patent for the PRO system using concentrated seawater and treated sewage water, acquired by Kyowakiden Industry Co., Ltd., can also use river water instead of treated sewage. Figure 4 shows the configuration of a PRO system that recovers energy by using turbines. This system has excellent advantages for installation in existing plants. This is because the seawater desalination plant and the PRO system can be operated independently. Highly efficient energy recovery can be achieved with a recovery rate of 92% by coaxially connecting a water-turbine and a high-pressure pump for desalination of seawater instead of a turbine for power generation.

Figure 5 shows the configuration of a PRO recovering energy system that uses PX^®^. In this system, the seawater desalination plant uses a two-stage PX^®^. However, when treated sewage water is used as the freshwater source for PRO, the treated sewage water is mixed with the seawater supplied by the seawater desalination plant. In some countries, including Japan, such a system cannot be used because sewage water should not be mixed with drinking water. However, other countries, like the USA and Australia, reuse treated wastewater and can be expected to take advantage of this system. In areas that do not reuse treated sewage water for drinking water, RO-treated water using this system may not be used as drinking water but could still be used as water for sprinkling. When introducing PRO to an existing seawater desalination plant, the system shown in Figure 4 can be widely used from the perspective of mixing RO-treated water with treated sewage water.

#### 2.1.2. PRO Membranes

PRO membranes require high permeability, pressure tightness, and anti-fouling potential [31]. In the early development of PRO membranes, conventional RO membranes were considered. However, the water permeability of conventional RO membranes was too low. Therefore, membranes specifically for PRO systems have been developed. Hollow fiber membranes with high water permeability are currently available in the market [6]. The development of flat sheet membranes for PRO systems has been slow at the commercial level. However, as the development of hollow fiber membranes for PRO systems progressed, better water permeability was obtained. However, film surface and internal membrane fouling remains a problem.

Many pretreatment approaches for fouling suppression have been studied [32,33]. In addition, researchers are trying to make breakthroughs in membrane development. The authors greatly look forward to membrane development leading to inexpensive membranes. Many researchers are working on lab-scale PRO membranes. Liu et al. used one-step phase inversion technology to study the improvement of fouling resistance by introducing silver nanoparticles (AgNPs) into the membrane substrate [34]. The AgNP is an effective biocide against various aquatic microorganisms, and nanocomposite membranes with AgNP exhibit better permeability and separation performance, in addition to the anti-fouling effect, and improvements in mechanical strength and thermal stability have been also been reported. In addition, new materials for PRO membranes are being actively developed. Zhang et al. developed a thin Polyethersulfone (PES) composite hollow fiber membrane [35]. The power density of this membrane was 24.3 W/m^2^ when 1 M NaCl and DI water were evaluated at 0.2 MPa for FO operation. Li et al. developed a hyperbranched polyglycerol-grafted polyether sulfone hollow fiber membrane [36]. The membrane was evaluated for its anti-fouling effect from the adsorption test with *Escherichia coli* of bovine serum albumin. This membrane was washed to obtain a high wash recovery of up to 94%.

The development of membranes using carbon nanotubes (CNTs) is also in progress. Fan et al. reported the construction of reduced graphene oxide on CNT hollow fiber substrates via electrophoretic deposition coupled with a chemical-reduction process for membranes on nanocarbon substrates [37]. This membrane has improved the permeability and ion selectivity by constructing a graphene active layer. In the FO test with 0.5 M NaCl and DI water, a water flux of 22.6 LMH was obtained, which is equivalent to 3.3 times the permeability of commercially available membranes. Tang et al. developed a thin film composite (TFC) membrane with a single-walled carbon nanotube (SWCNT) intermediate layer [38]. A membrane in which an intermediate layer is formed through a strong π-π interaction was prepared with SWCNTs with no functional group. The TFC membrane with the optimum SWCNT intermediate layer had a permeability of 3.3 L m^−2^ h^−1^ bar^−1^ in the RO test using 0.5 M NaCl. The TFC membrane with the optimum SWCNT intermediate layer had a high permeability of 62.8 L m^−2^ h^−1^ in the FO test with 1 M NaCl and DI water. This is because the TFC membrane with the SWCNT intermediate layer achieved higher permeability due to the significant reduction in structural parameters and the significant suppression of internal concentration polarization in the support layer.

Bench-scale PRO membrane manufacturers include Toray, AQUAPORIN, and Polyfera. Toray manufactures 8-inch spiral PRO membrane modules at Toray Chemical Korea, a group company [39]. The A, B, and S values of this film are 1.97 L h m^2^-h-bar, 0.619 L/m^2^-h, and 0.713 mm, respectively. However, this 8-inch spiral PRO membrane had a low power density, as compared to the performance of Toyobo’s hollow fiber PRO membrane module. AQUAPORIN manufactures composite polyamide hollow fiber membranes with aquaporins (proteins) in three types of modules with membrane areas of 0.6, 2.3, and 13.8 m^2^. AQUAPORIN’s membrane is recommended for use in the FO process used in semiconductor wastewater treatment. Porifera manufactures a novel flat membrane FO membrane module with a membrane area of 63 m^2^. FO membranes that can be used for concentration and wastewater treatment are manufactured by many manufacturers. Osmotic Engineering (UK), Forward Water Technologies (Canada), EDERNA, and Trevi Systems have reported their own FO systems [40].

Trends in commercial PRO membranes and manufacturers are of great interest to users and researchers. Currently, the only commercial-use PRO membrane is manufactured by Toyobo. Toyobo announced that their membrane was used in the demonstration test of PRO (FO), which was started by the Danish venture company Salt Power Aps in 2018 [41]. At the demonstration site, warm underground salt water was used as a local heating system. The PRO demonstration test used this salt water and fresh water to generate 20 kW of power.

The development of commercially available PRO membranes is gradually expanding to other manufacturers. The membranes that some manufacturers term as FO membranes have low-pressure resistance and are therefore not applicable for the PRO process. However, FO and PRO membranes are manufactured by a similar method. The authors expect that the expansion of the PRO membrane market will increase upon using the FO membrane.

### 2.2. Current Status of Existing Seawater Desalination Plants

As of 2019, there were 15,906 seawater desalination plants in operation [42]. The total water production of all seawater desalination plants was 95.37 million m^3^/d. In the 1980s, 84% of the world’s desalinated water was produced by the MSF and MED methods. The use of the RO method has increased since the 1980s, and, in the 2000s, the amount of desalinated water produced by thermal technology and RO was approximately 11.6 million m^3^/d and 11.4 million m^3^/d, respectively. Figure 6 shows the change in the size of RO plants for desalination and reclamation [43,44]. Since 2010, the amount of seawater desalinated in plants using RO has reached 100,000 m^3^/d. The size of plants using Seawater RO in 2020 has reached the mega-ton scale, and it is believed that the number of plants is further increasing. However, issues stemming from the discharge of chemicals and high-concentration salt water on marine organisms from seawater desalination plants should also be considered. Although there are a few studies that deny the effects on marine pollution, these are not long-term studies. The discharge of concentrated seawater from seawater desalination plants into the sea is of concern due to the environmental stress of high temperature, high salinity, and chemicals, which are known to affect marine organisms. However, there is no globally unified protocol for treating concentrated seawater, considering the five established treatment methods currently in use:1.Directly discharged into the sea or rivers;2.Discharged into the existing sewage system;3.Discharged into the evaporation area (Evaporation Pond);4.Discharged into deep wells;5.Zero Liquid Discharge (ZLD).

Many seawater desalination plants directly discharge concentrated seawater into seas or rivers. In this case, it has been reported that the local increase in salt content and the diffusion of concentrated seawater can reduce environmental stress; however, this method is not completely appropriate. Some countries and regions have regulations for concentrated seawater discharged from seawater desalination plants. These regulations often include a compliance point and critical concentration for discharge. Table 2 shows the critical concentration and compliance points for each region [45].

Wang et al. reported on marine pollution from seawater desalination plants in China [46]. The Qingdao Baifa and BEWG Aqualyng Seawater Desalination Projects are located in the Bohai Economic Rim. Large amounts of concentrated salt water, corrosion products, and chemical cleaning agents have caused marine pollution in this region, owing to inadequate water circulation. The Water Law was established in the People’s Republic of China, in 2016, for reducing marine pollution in China.

Marine pollution is a serious problem in the Middle East. Here, concentrated seawater tends to accumulate because all seawater desalination plants are located on the coasts of the Red Sea, the Gulf of Aqaba, and the Arabian Gulf. The amount of concentrated seawater discharged in the Red Sea is reported to have increased from 6.4 Mm^3^/d in 1996 to 142 Mm^3^/d in 2018. Ozair et al. reported that the temperature, salt content, dissolved oxygen, and phosphate contents in the Red Sea have also changed [47]. The development of the seawater desalination industry is important in the Middle East; however, environmental monitoring and stringent regulations are required.

Kelaher et al. reported the effects of discharging high-concentration salt water from the Sydney seawater desalination plant in the sea, along with the results of a seven-year survey of the habitat of living organisms in the region [48]. The outlet for concentrated seawater from the Sydney Desalination Plant was located on a reef approximately 300 m offshore at a depth of 25 m. The operating plant discharged an average amount of 342 ML/d of concentrated seawater with an average temperature and conductivity of 20.1 °C and 76,608 mS cm^−1^, respectively. A high-pressure diffuser was installed at the drain to increase the mixing ratio with seawater. Modeling and empirical data showed that the diffuser effectively mixed seawater and concentrated seawater. At 100 m from the discharge port, the difference in salinity from the surrounding seawater is less than 0.1 psμ (0.01 *w*/*w* %). The number of fish around the discharge port increased by 279% after the diffuser was implemented at the plant. This study also evaluated the changes that occurred when the discharge of concentrated seawater was voluntarily stopped; through increasing salinity or other changes, the concentrated seawater was observed to have certain environmental impacts.

Catastrophic marine pollution has not been reported since the introduction of the seawater desalination plant. However, local changes in salinity and temperature have been reported. The author proposes that this solution should be strengthened here before serious marine pollution is caused by seawater desalination plants. Upon introducing the PRO system, the concentration of brine discharged from the plant will be lowered. The PRO system can dilute the brine salinity of 7.0 *w*/*w* % from the seawater desalination plant to 3.9 *w*/*w* %, thus decreasing the environmental stress.

### 2.3. Comparison of Current Power Generation Costs by PRO and Electricity Charges

Kyowakiden Industry Co., Ltd., reported the power generation costs when PRO was installed in a mega-ton-scale seawater desalination plant [24]. The power generation costs for PRO have not changed significantly in the past few years as PRO membrane costs have remained steady. Table 3 shows the power generation costs by PRO. Freshwater volumes at 0.1 and 1 million m^3^/d of concentrated seawater are 0.079 and 0.79 million m^3^/d, respectively. The salinity of the discharged water in these cases is 3.9 *w*/*w* %. Thus, the power generation costs for 0.1 and 1 million m^3^/d of concentrated seawater are 0.28 and 0.19 USD/kWh, respectively. Subsequently, the running costs for 0.1 and 1 million m^3^/d of concentrated seawater are 0.11 and 0.08 USD/kWh, respectively. The estimated mega-ton-scale equipment costs are divided into membrane, equipment, civil engineering, and labor costs, which amount to 49%, 31%, 12%, and 8% of the total cost, respectively. The membrane cost in this case is currently 3200 USD/kWh. If this decreased to 550 USD/kWh, the membrane cost would be reduced to 18%.

Statkraft estimated the cost of osmotic power generation from seawater and fresh water to be 0.144 USD/kWh, which has been described in Section 2.1. Figure 7 shows the relationship between the seawater desalination plants using the RO method and industrial electricity charges (USD/kWh) [42]. Industrial electricity charges in each country are determined based on the values of each electric power company. Australia is the most promising market in terms of these costs as compared to industrial electricity charges. If the Statkraft estimation of 0.144 USD/kWh can be realized, PRO could be expected to be introduced in many countries including Malaysia, Singapore, the USA, the Netherlands, and Israel. In other words, PRO-based power plants using seawater and fresh water are promising sustainable energy sources.

The total amount of concentrated seawater produced globally by seawater desalination plants was estimated to be 182 Mm^3^/d. The total amount of concentrated seawater was calculated from the amount of drinking water produced by seawater desalination plants around the world and the recovery rate for the technology used in each plant. The amount of drinking water produced by seawater desalination plants worldwide is 95.4 Mm^3^/d [42]. The total power generation when PRO is introduced for each seawater desalination technology is 1.03. TWh/year. Table 4 shows the amount of power generated using PRO for each seawater desalination technology. In addition, the total power generation from seawater and freshwater PRO, using river water, is 650 TWh/year. Table 5 shows the total power generation by PRO from natural seawater and fresh water calculated as per the flow rate of rivers worldwide. The total sustainable water discharge was 124,600 Mm^3^/d [49]. The estimated pump efficiencies and turbine power generation efficiencies were 90% and 92%, respectively. The permeation rate of fresh water with respect to salt water was set to 65% and 70%, with saltwater concentrations of 7.0 *w*/*w* % and 3.5 *w*/*w* %, respectively. The total amount of generated power calculated by PRO, using seawater and fresh water, based on the total river water flow rate, was 877 GW. This energy source is an unused resource. The power generation by PRO with seawater and fresh water is one solution to realize a sustainable society.

International cooperation is vital for the popularization of PRO. In many countries, renewable energy sources are often subsidized by the government. PRO has the potential to be sufficiently competitive in many countries if subvention is introduced. This is because running costs for PRO are cheaper than industrial electricity charges in many countries and regions. One of the factors driving up power generation costs is the cost of the membranes. The membranes account for more than 40% of equipment costs. This is because PRO membranes are not widely manufactured, considering their limited use. If PRO membrane costs were approximately equal to those of the current RO membrane, the power generation and running costs of 1 Mm^3^/d would be 0.08 and 0.03 USD/kWh, respectively. Considering the current global environment, there is an urgent need to solve environmental issues such as the reduction of CO_2_ emissions. Therefore, the widespread use of PRO as a renewable energy source will expand the market for PRO membranes, allowing for PRO membranes to be provided at lower costs, similar to RO membranes.

## 3. Conclusions

To realize a sustainable society, it is necessary to increase the utilization of renewable energy; availability of a variety of renewable energy sources is also vital. Kyowakiden Industry Co., Ltd., recommended a PRO system that uses concentrated seawater and fresh water to reduce the energy consumption of the seawater desalination process by 10%. In addition, this PRO system, using natural seawater and fresh water, can be an energy source that contributes to the realization of a sustainable society, providing a better membrane module is developed. One of the better membrane module performances generates a power density of 6.5 W/m^2^ or higher [7].

In this paper, the technical details of the PRO system, the current status of the existing seawater desalination plants, and the power generation costs of the PRO system are summarized. The PRO system can be operated continuously for more than one year following a performance evaluation test. Furthermore, the scale of existing seawater desalination plants is also increasing. The number of mega-ton-scale seawater desalination plants is expected to increase in the 2020s. The current cost of the PRO membrane is 3200 USD/module; the power generation cost for PRO, using concentrated seawater of 1,000,000 cubic meter per day (CMD), is 0.19 USD/kWh. If the PRO membrane cost is 550 USD/module, the power generation cost for PRO at the mega-ton scale is 0.09 USD/kWh. The PRO system is fully applicable for large-scale seawater desalination plants. The power-generation cost of the PRO system is sufficiently competitive at the mega-ton scale, as compared to current industrial electricity charges. The total power generation capacity of PRO using concentrated seawater and fresh water is 3 GW. The total power generation capacity of PRO, using natural seawater and fresh water, is 1781 GW. However, the net output power is approximately 50% of these values.

Kyowakiden Industry Co., Ltd., determined that the PRO system should be commercialized, owing to ongoing energy issues and the technical state-of-the-art of the PRO system. Therefore, preparations are underway to carry out a large-scale demonstration at a seawater desalination plant, to facilitate the global introduction of PRO systems for the realization of a sustainable society.

## Figures and Tables

**Figure 1 membranes-11-00069-f001:**
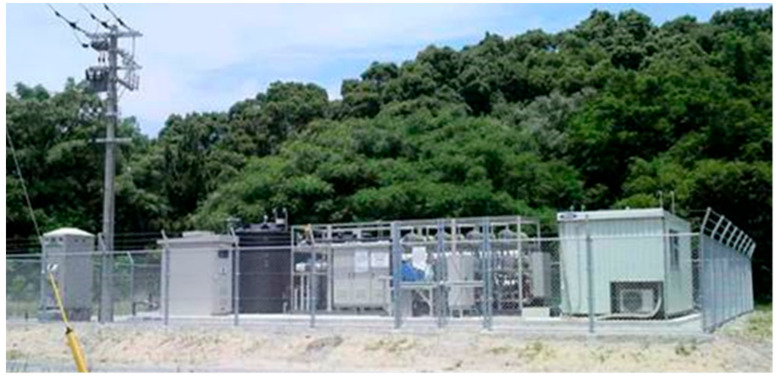
Photograph of the demonstration test site.

**Figure 2 membranes-11-00069-f002:**
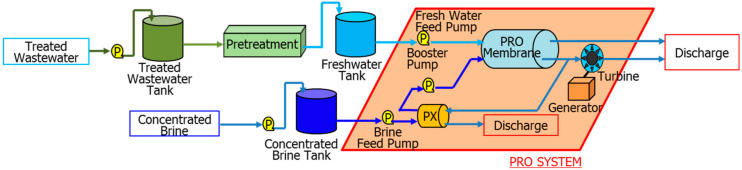
Overview of a long-term demonstration test site. PRO, pressure retarded osmosis.

**Figure 3 membranes-11-00069-f003:**
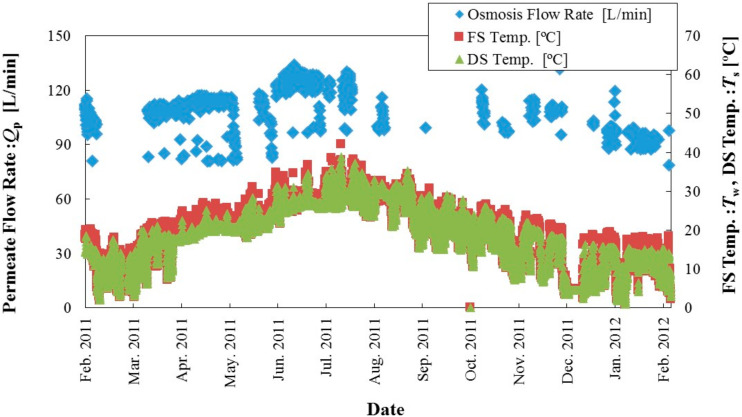
Time course of permeate flow rate and temperatures at our PRO prototype plant [24].

**Figure 4 membranes-11-00069-f004:**
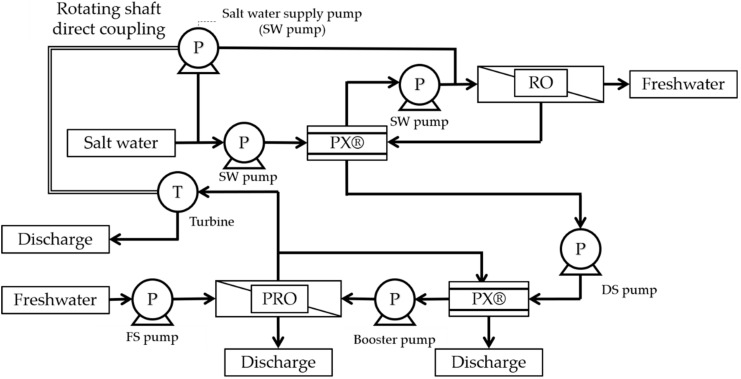
Configuration of the PRO system, in case of turbine use for energy recovery. PX^®^, Pressure Exchanger^®^.

**Figure 5 membranes-11-00069-f005:**
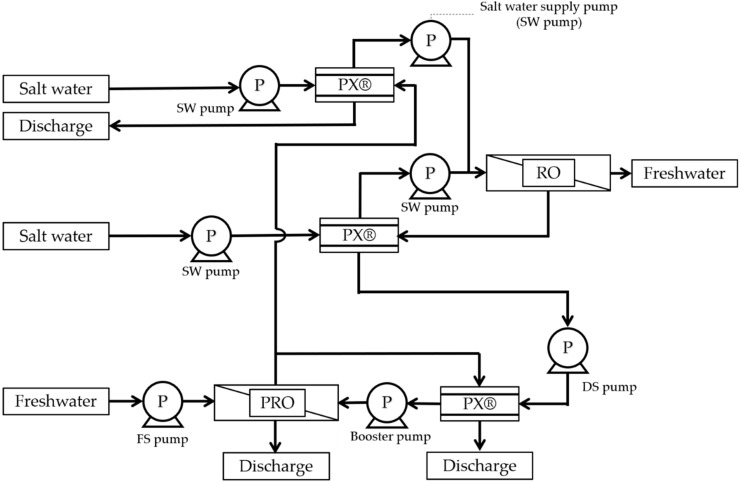
Schematic demonstrating the configuration of recovering energy by PX^®^ of the PRO system.

**Figure 6 membranes-11-00069-f006:**
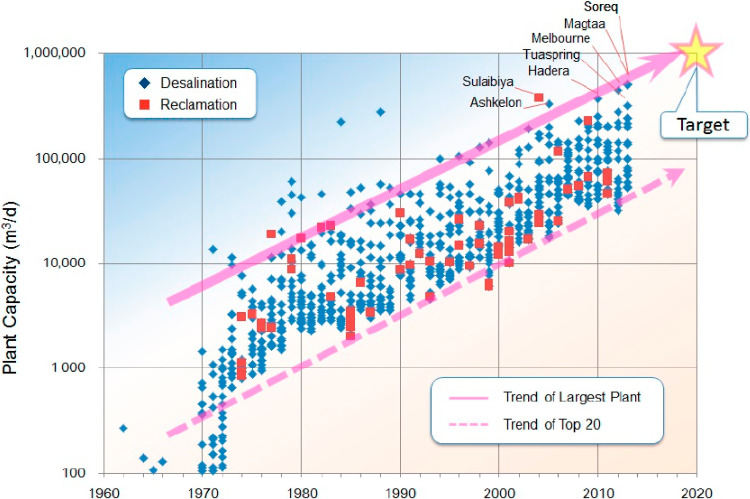
Change in the size of RO (reverse osmosis) plants for desalination and reclamation [43,44].

**Figure 7 membranes-11-00069-f007:**
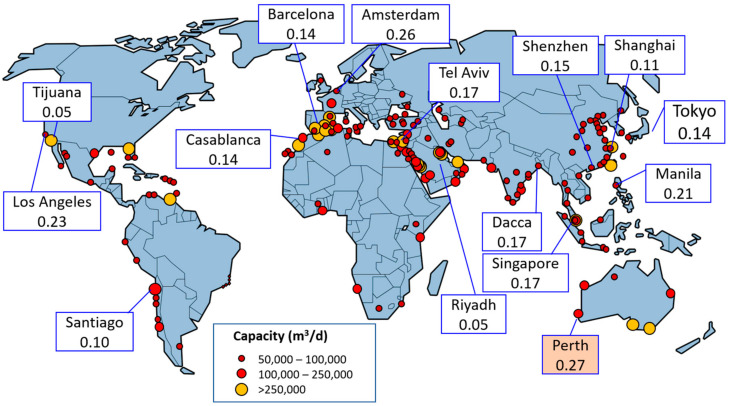
Relationship between seawater desalination plants using the RO method and industrial electricity charges (USD/kWh) worldwide.

**Table 1 membranes-11-00069-t001:** Materials used at a long-term demonstration test site.

	Item	Specification	Origin
Raw water	Concentrated brine	460 m^3^/d	Mamizu Pia (Desalination Plant)
	Treated wastewater	420 m^3^/d	Wajiro Wastewater Treatment Center
Freshwater pretreatment	UF	RS50-S8 (8 inches)	NITTO (Reused membrane)
	Low-pressure RO	ES20B-D8 (8 inches)	NITTO (Reused membrane)
PRO membrane	Hollow fiber	4 ports 10 inches × 8	TOYOBO
Water-turbine generator	Pelton	Power 7.7 kW	Canada, Japan

RO, reverse osmosis.

**Table 2 membranes-11-00069-t002:** Critical concentration and compliance points for each region [45].

Region/Authority	Salinity Limit	Compliance Point (Relative to Discharge)
US EPA	Increment ≤ 4 ppt	
Carlsbad, CA	Absolute ≤ 40 ppt	1000 ft.
Huntington Beach, CA	Absolute ≤ 40 ppt salinity (expressed as discharge dilution ratio of 7.5:1)	1000 ft.
Western Australia guidelines	Increment < 5%	
Oakajee Port, Western Australia	Increment ≤ 1 ppt	
Perth, Australia/Western AustraliaEPA	Increment ≤ 1.2 ppt at 50 m and ≤ 0.8 ppt at 1000 m	50 and 1000 m
Sydney, Australia	Increment ≤ 1 ppt	50–75 m
Gold Coast, Australia	Increment ≤ 2 ppt	120 m
Okinawa, Japan	Increment ≤ 1 ppt	Mixing zone boundary
Abu Dhabi	Increment ≤ 5%	Mixing zone boundary
Oman	Increment ≤ 2 ppt	300 m

**Table 3 membranes-11-00069-t003:** Power-generation costs for PRO.

	PRO System Size	Mega-Ton Water PRO System Size	Mega-Ton Water PRO System Size (Future Price of the PRO Membrane)
Draw side	Brine (Concentration 7%)		
Concentrated brine (CMD)	100,000	1,000,000	1,000,000
PRO membrane cost (USD/module)	4100	3200	550 *
Pump efficiency (%)	85		
Turbine efficiency (%)	88	92 **	92 **
Power density (W/m^2^)	12		
Net output power (kW)	1100	12,000	12,000
Facility redemption (USD/kWh)	0.17	0.11	0.06
Running cost (USD/kWh)	0.11	0.08	0.03
Generation cost (USD/kWh)	0.28	0.19	0.09

* Price similar to RO membrane module. ** Power recovery directly connected to the shaft. Conversion: 1 USD = 110 JPY.

**Table 4 membranes-11-00069-t004:** Amount of generated power using PRO for each seawater desalination technology.

	RO	MSF	MED	ED	Other	Total	
Seawater desalination plant capacity [42]	64	18	8.5	2.5	2.37	95.4	million m^3^/d
Recovery [42]	42	22	25	86	4		%
Brine capacity	88	64	26	0.4	3.6	182	million m^3^/d
Brine concentration	6.03	4.5	4.7	25	5.8		*w*/*w* %
Osmotic pressure	5.1	3.8	4.0	21	4.9		MPa
PRO DS pressure	2.5	1.8	1.9	6 *	2.4		MPa
Permeation rate	65						%
Power generation	0.57	0.32	0.14	0.007	0.03	1.03	TWh/year
	1.6	0.9	0.4	0.019	0.08	3	GW
Net Output Power	0.35	0.16	0.07	0.006	0.02	0.57	TWh/year
	1.0	0.4	0.2	0.016	0.05	1.7	GW

* Considered as proof of the pressure of the membrane. MSF, multi-stage flash (MSF); MED, multi-effect desalination (MED).

**Table 5 membranes-11-00069-t005:** Total power generation by PRO for natural seawater and fresh water, calculated by the flow rate of rivers worldwide.

Total Sustainable Water Defined as Discharge	124,600	Million m^3^/d
Seawater concentration	3.5	*w*/*w* %
Osmotic pressure	3.0	MPa
PRO DS pressure	1.5	MPa
Permeation rate	70	%
Power generation	650	TWh/year
	1781	GW
Net output power	320	TWh/year
	877	GW

## Data Availability

Not applicable.

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
