# Peer review of "Commercial Pressure Retarded Osmosis Systems for Seawater Desalination Plants"

_membranes, 2021, doi:10.3390/membranes11010069_

Round 1
Reviewer 1 Report
Very impressive work with lots of commercial opportunity. The paper requires more detail on the results obtained in the project. There must a lot of invaluable data from the pilot plant that could help the reader realise the potential based on practical results.
Section 2.1 mentions many different types of membrane but does not report on the performance of the TOYOBO membranes used in this project. Can you provide some detailed results on the performance of the TOYOBO membranes used in this project. A couple of tables/figures here on flux and fouling and costs would help significantly.
Section 2.2 needs to make a connection with the developed PRO system. It is absolutely correct that discharged brines from desalination plants cause negative environmental impact but how does the PRO system help this effect?
The total quantity of salt in the discharge does not change. The PRO system produces two more diluted discharge streams but the total quantity of TDS discharged to the sea/river is the same. Have you considered using the PRO discharge to dilute the seawater and use that as feed to the desalination plant? This would reduce the energy consumption of the desalination plant.
Line 382: "Membranes account for more than 40% of equipment costs." Can you elaborate? What fractions of the total cost are the rest of the equipment/instrumentation/control system. Also, how much more are the PRO membranes' cost compared to RO membranes?
The Conclusions section needs much more detailed information/data about the specific project this paper is reporting on. Include some figures on the efficiencies achieved in the project and what further steps you are planning to take in order to commercialise the technology. Same goes for the Abstract.
Typo in Line 69
Line 182: flat "sheet" membranes
Table 4: Units are missing for the osmotic pressure
Author Response
Very impressive work with lots of commercial opportunity. The paper requires more detail on the results obtained in the project. There must a lot of invaluable data from the pilot plant that could help the reader realise the potential based on practical results.
We appreciate your kind advice and comments. Thanks to your valuable comments, we believe that the manuscript has been improved satisfactorily.
Section 2.1 mentions many different types of membrane but does not report on the performance of the TOYOBO membranes used in this project. Can you provide some detailed results on the performance of the TOYOBO membranes used in this project. A couple of tables/figures here on flux and fouling and costs would help significantly.
We are sorry that this description is unclear. We should not provide the full specification. Text was revised. And Figure was added.
(line 120-125)
Texts were changed from
“This test site includes eight 4-way 10-inch Toyobo CTA hollow fiber membrane modules, UF membranes, low-pressure RO membranes for advanced treatment of sewage-treated water (Pretreatment of feed solution), and a Pelton turbine generator. Then, the operating status was acquired using a 24-hour automatic control and monitoring system.”
to
“The demonstration test site includes eight 4-way 10-inch Toyobo CTA hollow fiber membrane modules, UF membranes, low-pressure RO membranes for advanced treatment of sewage-treated water (Pretreatment of feed solution), and a Pelton turbine generator. The hollow fiber membrane has an outer diameter of 0.2 mm, inner diameter of 0.1 mm and the length is 1.3 m. Operating status was acquired using a 24-hour automatic control and monitoring system.”
(line 126-133)
Texts were changed from
“Through this demonstration test, Kyowakiden Industry Co., Ltd. obtained a high amount of power generation (Power density: 10 W/m2) and temperature changes due to seasonal variation (Draw solution: 2–38 , Feed solution: 1–40 ) in the performance over the year.”
to
“Through this test, Kyowakiden Industry Co., Ltd. obtained a large amount of power generation (Power density: 10 W/m2) and temperature changes due to seasonal variation (Draw solution: 2–38 , Feed solution: 1–40 ) in the performance over the year. Figure 3. shows time course of permeate flow rate and temperatures at the PRO prototype plant [24]. The permeate flow through the membrane is temperature-dependent, similar to the RO membrane method. The same membrane module was used consecutively in this test. The permeate flow rate did not decrease in the continuous one year test.”
Section 2.2 needs to make a connection with the developed PRO system. It is absolutely correct that discharged brines from desalination plants cause negative environmental impact but how does the PRO system help this effect?
Thank you for your kind comment. The negative environmental impact of brine is a concern near the outlet. At points away from the outlet, the negative environmental impact of brine is low. Because brine is diluted with seawater. However, since there is always brine near the outlet, the negative environmental impact will be prolonged. In the PRO system is introduced, the concentration of brine discharged from the outlet is low. The brine salinity 7.0 w/w% from the seawater desalination plant is diluted to a salinity 3.9 w/w% by the PRO system. The lower the negative environmental impact with the lower the concentration of discharge water. Text was revised.
(line 334)
Texts were added
“Upon introducing the PRO system, the concentration of brine discharged from the plant will be lowered. The PRO system can dilute the brine salinity of 7.0 w/w% from the seawater desalination plant to 3.9 w/w%, thus decreasing the environmental stress.”
The total quantity of salt in the discharge does not change. The PRO system produces two more diluted discharge streams but the total quantity of TDS discharged to the sea/river is the same. Have you considered using the PRO discharge to dilute the seawater and use that as feed to the desalination plant? This would reduce the energy consumption of the desalination plant.
Thank you for your advice. The method proposed by the reviewer reduce the energy consumption of seawater desalination plants. However, producing drinking water from the river consumes less energy. The purpose of our PRO system is to recover energy from the currently discharged concentrated seawater. The high energy of discharge concentrated seawater is not being used.
Line 382: "Membranes account for more than 40% of equipment costs." Can you elaborate? What fractions of the total cost are the rest of the equipment/instrumentation/control system. Also, how much more are the PRO membranes' cost compared to RO membranes?
We are sorry that this description is unclear. We revised the texts.
(line 348-352)
Texts were changed
“The estimated mega-ton scale equipment costs are divided into membrane, equipment, civil engineering, and labor costs, which amount to 49, 31, 12, and 9% of the total cost, respectively. The membrane cost in this case is currently 3200 USD/kWh. If this decreased to 550 USD/kWh, the membrane cost would be reduced to 18%.”
The Conclusions section needs much more detailed information/data about the specific project this paper is reporting on. Include some figures on the efficiencies achieved in the project and what further steps you are planning to take in order to commercialise the technology. Same goes for the Abstract.
Thank you for your kind comment. The abstracts and conclusions were not clear. Text was revised.
(line 11-14)
Texts were changed
“PRO requires salt water and fresh water, both of which can be found at seawater desalination plants. The total power generation capacity of PRO using concentrated seawater and freshwater is 3 GW. Massive amount of energy is required for seawater desalination, therefore, the introduction of renewable energy should be prioritized.”
(line 411-415)
Texts were changed
“In addition, the PRO system using natural seawater and freshwater can be an energy source that contributes to the realization of a sustainable society, providing a better membrane module is developed. One of the better membrane module performances generates a power density of 6.5 W/m2 or higher [7].”
(line 420-424)
Texts were changed
“The number of mega-ton-scale seawater desalination plants is expected to increase in the 2020s. The current cost of the PRO membrane is 3200 USD/module, the power generation cost for PRO using concentrated seawater of 1,000,000 CMD is 0.19 USD/kWh. If the PRO membrane cost is 550 USD/module, the power generation cost for PRO at the mega-ton scale is 0.09 USD/kWh.”
Typo in Line 69
Line 182: flat "sheet" membranes
Table 4: Units are missing for the osmotic pressure
We are very sorry for such a careless point. Text and Table were revised.
(line 67-69)
Texts were changed
“In addition, Flowserver’s Calder DWEER and Fluid Equipment, Development Company's (FEDCO) HPB Turbocharger are used [20].”
(line 189-190)
Texts were changed
“The development of flat sheet membranes for PRO systems has been slow at the commercial level.”
(line 389)
Table was changed.
Reviewer 2 Report
The manuscript is all about the utilization of PRO system in seawater desalination plants. Overall, this manuscript is well organized and give a lot of information on the real system. The manuscript can be accepted after minor revision. The following are my concerns:
- The PRO membraned used in this work is from TOYOBO. Is it possible to include the physical and chemical property of the PRO membrane used in this work? (pore size;type of membrane , thin-film or not, etc)
- Authors provide the process flow diagram in Figure 3 & 4; is it possible to transform this figure to more detail - A piping and instrumentation diagram, including the temperature control or flow control and other valves that are presented in the process.
Author Response
The manuscript is all about the utilization of PRO system in seawater desalination plants. Overall, this manuscript is well organized and give a lot of information on the real system. The manuscript can be accepted after minor revision. The following are my concerns:
We appreciate your kind advice and comments. Thanks to your valuable comments, we believe that the manuscript has been improved satisfactorily.
The PRO membraned used in this work is from TOYOBO. Is it possible to include the physical and chemical property of the PRO membrane used in this work? (pore size;type of membrane , thin-film or not, etc)
We are sorry that this description is unclear. We should not provide the full specification. Text was revised.
(line 120-125)
Texts were changed from
“This test site includes eight 4-way 10-inch Toyobo CTA hollow fiber membrane modules, UF membranes, low-pressure RO membranes for advanced treatment of sewage-treated water (Pretreatment of feed solution), and a Pelton turbine generator. Then, the operating status was acquired using a 24-hour automatic control and monitoring system.”
to
“The demonstration test site includes eight 4-way 10-inch Toyobo CTA hollow fiber membrane modules, UF membranes, low-pressure RO membranes for advanced treatment of sewage-treated water (Pretreatment of feed solution), and a Pelton turbine generator. The hollow fiber membrane has an outer diameter of 0.2 mm, inner diameter of 0.1 mm and the length is 1.3 m. Operating status was acquired using a 24-hour automatic control and monitoring system.”
Authors provide the process flow diagram in Figure 3 & 4; is it possible to transform this figure to more detail - A piping and instrumentation diagram, including the temperature control or flow control and other valves that are presented in the process.
Figures 3 and 4 are configurations. For this reason, these figures do not include the temperature control or flow control and other valves. However, the figure title was incorrect. Text was revised.
(line 164)
Texts were changed from
“Figure 3. Concept of PRO system, in case of use of turbine for energy recovery.”
to
“Figure 4.  Configuration of the PRO system, in case of turbine use for energy recovery.”
(line 164)
Texts were changed from
“Figure 4.  Schematic demonstrating the concept of recovering energy by PX of the PRO system.”
to
“Figure 5. Schematic demonstrating the configuration of recovering energy by PX of the PRO system.”

Reviewer 3 Report
This problem is directly relevant for journal scope. The manuscript follows the formal regulations of journal. The manuscript seems the mixture of a case study and review work.
I suggest to accept for publication this work after major revision.
Remarks, suggestions, questions
- I suggest the following structure and complete editing: Introduction, Materials and Methods, Results and Discussion. The theory must be clearly separated from the communication of the results.
- Please add information about mass- and component balance.
- I suggest the comprehensive review of English literature.
- Please add the reference(s) of Table 4. Please review the whole manuscript, all industrial data should be referenced.
Author Response
This problem is directly relevant for journal scope. The manuscript follows the formal regulations of journal. The manuscript seems the mixture of a case study and review work. I suggest to accept for publication this work after major revision. Remarks, suggestions, questions
Thank you for your kind advice and comments. We reconsidered various points and revised the text. We believe that the manuscript has been improved satisfactorily.
I suggest the following structure and complete editing: Introduction, Materials and Methods, Results and Discussion. The theory must be clearly separated from the communication of the results.
Thank you for your kind comment. However, since our paper is structured more like a discussion. This may be a little difficult to achieve. We referred to the demonstration test results in 2.1.1. In this paper, it is described in "Results and Discussion" to clarify the research trend of PRO system. However, this paper does not simply introduce these references. We evaluate the feasibility of the PRO system from the references and our estimation results.
Please add information about mass- and component balance.
Thank you for your kind comment. There was no description of mass balance in the power generation cost estimation. Texts were added.
(line 343-345)
Texts were added.
“Freshwater volumes at 0.1 and 1 million m3/d of concentrated seawater are 0.079 and 0.79 million m3/d, respectively. The salinity of the discharged water in these cases is 3.9 w/w%.”
I suggest the comprehensive review of English literature.
We have undergone a native check on the manuscripts we have already submitted. This manuscript was also natively checked.
Please add the reference(s) of Table 4. Please review the whole manuscript, all industrial data should be referenced.
We are sorry for such a careless point. The reference number in Table 4 was missing. We revised the Table 4.

Round 2
Reviewer 3 Report
Thank you very much for your kind comments and answers. I suggest the acceptance in this present form for publication.